DISCOVERY REPORT

# Multiple molecular events underlie stochastic switching between 2 heritable cell states in fungi

**Naomi Ziv** *, **Lucas R. Brenes**¤, **Alexander Johnson**\*

Department of Microbiology and Immunology, University of California, San Francisco, San Francisco, California, United States of America

¤ Current address: Biology Graduate Program, Massachusetts Institute of Technology, Cambridge, Massachusetts, United States of America
* nz375@nyu.edu (NZ); ajohnson@cgl.ucsf.edu (AJ)

**Data Availability Statement:** All data and scripts used for analysis are available through the Open

## Abstract

Eukaryotic transcriptional networks are often large and contain several levels of feedback regulation. Many of these networks have the ability to generate and maintain several distinct transcriptional states across multiple cell divisions and to switch between them. In certain instances, switching between cell states is stochastic, occurring in a small subset of cells of an isogenic population in a seemingly homogenous environment. Given the scarcity and unpredictability of switching in these cases, investigating the determining molecular events is challenging. White-opaque switching in the fungal species *Candida albicans* is an example of stably inherited cell states that are determined by a complex transcriptional network and can serve as an experimentally accessible model system to study characteristics important for stochastic cell fate switching in eukaryotes. In standard lab media, genetically identical cells maintain their cellular identity (either "white" or "opaque") through thousands of cell divisions, and switching between the states is rare and stochastic. By isolating populations of white or opaque cells, previous studies have elucidated the many differences between the 2 stable cell states and identified a set of transcriptional regulators needed for cell type switching and maintenance of the 2 cell types. Yet, little is known about the molecular events that determine the rare, stochastic switching events that occur in single cells. We use microfluidics combined with fluorescent reporters to directly observe rare switching events between the white and opaque states. We investigate the stochastic nature of switching by beginning with white cells and monitoring the activation of Wor1, a master regulator and marker for the opaque state, in single cells and throughout cell pedigrees. Our results indicate that switching requires 2 stochastic steps; first an event occurs that predisposes a lineage of cells to switch. In the second step, some, but not all, of those predisposed cells rapidly express high levels of Wor1 and commit to the opaque state. To further understand the rapid rise in Wor1, we used a synthetic inducible system in *Saccharomyces cerevisiae* into which a controllable *C. albicans* Wor1 and a reporter for its transcriptional control region have been introduced. We document that Wor1 positive autoregulation is highly cooperative (Hill coefficient > 3), leading to rapid activation and producing an "all or none" rather than a graded response. Taken together, our results suggest that reaching a threshold level of a

Science Framework, DOI: 10.17605/OSF.IO/6E9VZ (https://osf.io/6e9vz/).

**Funding:** This study was supported by NIH grant R01AI049187 (A.D.J.) and aided by a grant from The Jane Coffin Childs Memorial Fund for Medical Research (to N.Z.).The funders had no role in study design, data collection and analysis, decision to publish, or preparation of the manuscript.

**Competing interests:** The authors have declared that no competing interests exist.

master regulator is sufficient to drive cell type switching in single cells and that an earlier molecular event increases the probability of reaching that threshold in certain small lineages of cells. Quantitative molecular analysis of the white-opaque circuit can serve as a model for the general understanding of complex circuits.

## Introduction

Transcription circuits, defined as transcription regulators and the DNA *cis*-regulatory sequences they bind, control the expression of genes and thereby define cellular identity. Cellular identity is not static; gene expression programs change in response to external and internal stimuli. Certain transcriptional networks can produce distinct cell states that can be maintained through many cycles of cell division. The best characterized of such circuits come from microbes, such as the cI-Cro circuit of phage lambda [1], the control of lactose utilization in *Escherichia coli* [2], competence in *Bacillus subtilis* [3], and galactose utilization in *Saccharomyces cerevisiae* [4]. Such "bistability" has also been engineered in both prokaryotic and eukaryotic synthetic circuits [5–7] and is thought to underlie many instances of cell differentiation in multicellular organisms [8–10]. The production of 2 stable transcriptional states from a single genome depends on positive feedback regulation and nonlinearity (such as cooperativity) within the feedback circuit, which can convert graded inputs into discontinuous switch-like outputs [11–13]. The output of a given circuit "diagram," however, is entirely dependent on the physical parameters of the components, such as dissociation constants and protein concentrations; even slight variations in circuit architecture or parameters can produce distinct outputs [14]. Many questions remain concerning the functional properties and parameters of network motifs found in real biological systems, particularly those based on the large regulatory networks commonly found in eukaryotic organisms. A major challenge to experimentally investigating the functional role of the multiple feedback loops present in complex circuits is the limited ability to independently manipulate the different components. In order to understand the output of these large networks, it is necessary to integrate the analysis of individual motifs with an analysis of the network as a whole.

In some cases, switching between states depends on external signals, but in other cases switching between stable states appears stochastic, occurring in a subset of cells of an isogenic population in a seemingly homogenous environment. In microbes, stochastic phenotypic switching is often thought of in terms of adaptation to fluctuating environments [15,16], where there is a predicted optimal relationship between the frequency of stochastic switching and the frequency of environmental change. In pathogens, stochastic switching can create distinct subpopulations with different cell features, protecting against targeted host defense systems [17]. Stochastic switching also occurs during development to create cellular diversity [18]. For example, photoreceptor patterning in the fly retina is based on a choice between cell states following the stochastic expression of a single transcription factor [19]. Understanding stochastic phenotype switching at a molecular level is challenging, in particular, identifying the early, seemingly random, initiating events that determine which cells will undergo switching.

To study the mechanism of stochastic switching between 2 stable transcription states, we investigate white-opaque switching in *Candida albicans*, a common component of the human gut microbiota but also an opportunistic pathogen causing life-threatening bloodstream infections [20]. White and opaque cells differ in the expression of hundreds of genes resulting in drastic differences in cell morphology, metabolism, the ability to mate, and interactions with the immune system [21–25]. A complex transcriptional network controls white-opaque

switching, as many (at least 7) transcription factors are involved and are known to regulate one another, in a series of nested feedback loops [26–28]. The master regulator of white-opaque switching is Wor1, a transcription factor whose deletion completely blocks switching (locking cells in the white state) and whose ectopic expression in white cells converts them en masse to opaque cells, even in the absence of other critical regulators [26,29–31]. Wor1 is thought to positively regulate itself, as ectopic Wor1 expression activates expression from the endogenous Wor1 locus, which is necessary for maintaining the opaque cell state when ectopic expression is removed [31]. Wor1 is differently regulated between the 2 cell states, with 40-fold higher expression in opaque cells. In common with the other regulators of white-opaque switching, the Wor1 regulatory sequences (7 kb upstream control region and 2 kb 5′ UTR) are some of the longest in the *C. albicans* genome [23]. Its complexity is reminiscent of enhancers from multicellular organisms, even exhibiting the phase transitions described for some mammalian enhancers [32]. Wor1 is positioned in the center of a network of multiple positive feedback loops, binding both its own promoter as well as those of both activators and repressors of the opaque cell state [26,27].

In this work, we investigate the process of white-opaque switching at a single-cell level. To do so, we developed 2 complementary quantitative approaches to study the molecular mechanisms underlying switching. The first is the use of microfluidics and fluorescent microscopy to follow switching in *C. albicans*, allowing us to quantify the activation of the master regulator Wor1 in pedigrees where both switching and nonswitching cells are dividing. The second is the use of a synthetic inducible system in *S. cerevisiae* to investigate the function of white-opaque regulators and characterize specific regulatory interactions, such as Wor1 autoregulation. Our results reveal features of white-opaque switching that were unanticipated from previous "bulk" culture analyses and provide key insights into the mechanism behind stochastic switching.

## Results

### Observing white-opaque switching at a single-cell level

White-opaque switching has typically been studied by monitoring the formation of colony sectors after plating single cells on solid agar and allowing them to grow and form colonies of approximately $10^6$ cells (Fig 1A and 1B, and Table A in S1 File). A sector registers a switching event that took place in the past and reveals the stochastic nature of switching (e.g., sectors of different sizes in different colonies) but provides no insight into how the original cell that founded the sector differed from the surrounding cells and underwent switching. We set out to observe switching at a single-cell level to understand how this process occurs. Using switching competent *C. albicans* a/Δ diploid cells, we took advantage of microfluidic plates that contain specialized barrier traps (Fig 1C and see Methods). As the experiment begins, traps contain a few cells and, over time, these cells divide, eventually filling the trap. After approximately 12 hours, each trap holds approximately 400 cells. We continue imaging cells for an additional 12 hours, during which cells continue to divide in the trap, pushing excess cells out of the trap where they flow away. With this experimental setup, we are able to observe thousands of divisions in each trap and, by monitoring many traps, capture detailed instances of stochastic white-opaque switching, even though these events are rare.

Wor1 is a master regulator of white-opaque switching, and its expression is 40-fold higher in opaque cells compared to white cells [31]. We followed switching events in a strain homozygous for a Wor1-green fluorescent protein (GFP) fusion protein (Fig 1D and S1 Video). Microfluidic experiments that began with cells from white colonies did not have any visible

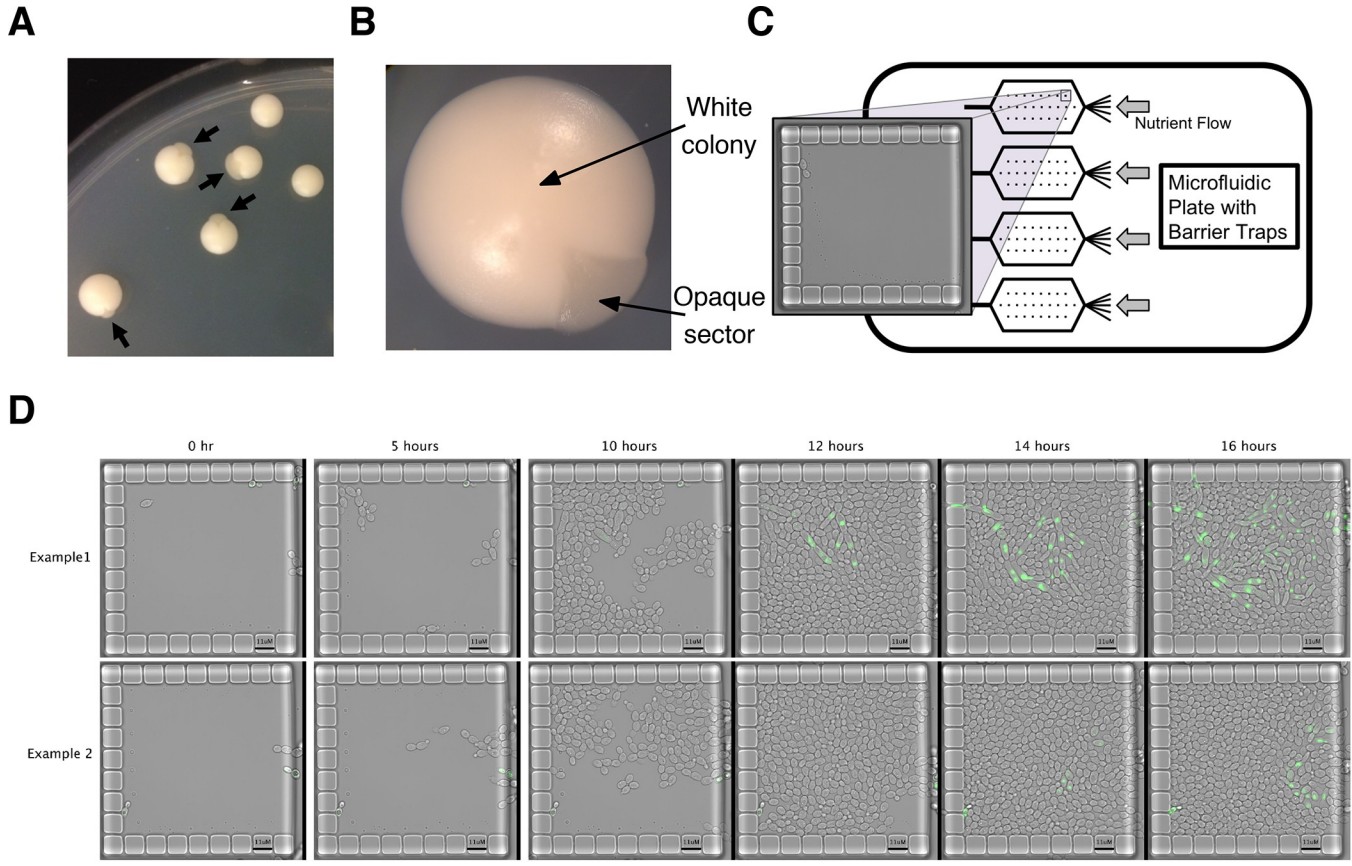

**Fig 1.** Monitoring white-opaque switching in *C. albicans* (A) Image of colonies produced by plating single white cells on solid agar. Arrows point to opaque sectors; each represents an independent switching event. This strain contains the Wor1-GFP fusion protein and hence exhibits higher than normal switching rates. (B) Close up image of a single white cell type colony with an opaque sector. (C) Schematic of microfluidic plates used to study white-opaque switching at a single-cell level. Each plate has 4 chambers; each chamber contains 104 barrier traps. (D) Two examples of microfluidic traps in which a switching event occurred. The strain contains the Wor1-GFP fusion protein, and cells initially entered the trap (from the right) as white cells. As cells complete switching to the opaque state, the Wor1-GFP fusion reaches high levels of expression.

GFP signal (Fig 1D), while those that began with cells from opaque colonies had uniformly high levels of GFP signal (Fig A in S1 File). Using this strain allowed for the unambiguous identification of switching events as well as the quantification of the increase in Wor1 expression as cells transition between the white and opaque states. Although switching is still relatively rare and stochastic, strains containing the Wor1-GFP fusion switch at higher rates than wild-type cells (Table A in S1 File), which enabled observation of many independent switching events. GFP is known to form dimers at high concentrations [33,34] and when we introduced the GFP A206K mutation (differentiated as mGFP below) to prevent dimerization [35], white-opaque switching reverted to wild-type levels (Table A in S1 File). By visualizing a large number of traps, we were able to observe several switching events in the Wor1-mGFP fusion strain. Apart from the difference in switching frequency, the switching behavior (as described below) of the Wor1-mGFP fusion strain appeared identical to the Wor1-GFP strain. Most of our data therefore was obtained from the elevated switching strain with the low-switching strain used to validate it. As predicted from previous bulk culture experiments, once the Wor1-GFP fusion reached a high level, it was faithfully passed on to descendent cells, preserving the opaque state.

## Observations on pseudohyphal growth and mitosis

Consistent with a previous report [36], we observed elongated cells reminiscent of pseudo-hyphae [37] as intermediates in some instances of white-opaque switching. Typically, a round mother cell buds an elongated cell and both subsequently activate Wor1 (Fig 2A and 2B and Fig B in S1 File). For the strain containing the Wor1-GFP fusion protein, approximately 14% of mothers and 80% of daughters in pairs of switching cells were longer than the 95 percentile of white cell lengths (Fig 2A). It is possible that pseudohyphal formation increases the probability of switching; however, it is not necessary or sufficient for switching as we observed pseudohyphal formation without switching and switching without pseudohyphal formation. In the majority of cases, descendants of initial switching cells resemble typical opaque cells (which are slightly more elongated than white cells) within a few cell divisions.

Although not a primary goal of this work, the nuclear localization of Wor1 allowed us to observe certain features of mitosis in switching cells. In a subset of cases, nuclear division occurred within the elongated daughter cell and not across the bud neck (Fig C in S1 File). Similar observations have been seen in filamentous forms of *C. albicans* [37]. In some of our examples, one of the 2 nuclei in the daughter cell returned to the mother cell and both cells continued dividing (Fig C in S1 File). In other examples, both nuclei remained in the elongated daughter forming polyploid cells (Fig C in S1 File). In these cases, the mother cell no longer divided. This distinctive mitosis pattern did not always occur during switching and, as a result, the majority of opaque cells are not polyploid. Although we were able to observe these deviations from conventional mitosis, they are not required for switching, as we did not observe them in all switching events. Furthermore, these atypical mitosis events are likely specific to *C. albicans* cells and do not represent a general feature.

## WOR1 activation is synchronous in mother–daughter pairs and is stereotypical across independent switching events

To quantify switching, we developed a custom semiautomated image analysis pipeline to measure Wor1-GFP (or Wor1-mGFP) levels in single cells over time (Fig D in S1 File and see Methods). Because Wor1 is nuclear localized, we used the average of the 300 highest intensity pixels per cell as a quantitative measure of Wor1 levels. During switching, Wor1 levels increase and then oscillate in concert with nuclear divisions similar to those observed in established opaque cells (Fig 2C). Comparing Wor1 activation in mother cells of independent switching events (taken from different traps and experiments) reveals a generally stereotypical response (Fig 2D and Fig E in S1 File) taking approximately 3 hours to reach maximum Wor1 levels. Consistent with the stochastic nature of switching, the starting time of different switching events did not depend on the time from the beginning of the experiment. Moreover, even the time between the birth of a switching cell and its switching varies from 1 cell to the next. For the set of examples shown in Fig 2D ($n = 21$), maximum Wor1 activation is reached between 5.4 to 17.2 hours after cell birth (mean = 9.9, standard deviation = 3.2).

As mentioned above, Wor1 activation occurs in dividing cells where both mother and daughter cells typically activate Wor1 simultaneously. This synchronous activation at times extends to the mother cells' previous daughter as well, leading to 4 cells (mother, 2 daughters, and 1 granddaughter) activating Wor1 simultaneously (Fig 2E and 2F). The synchrony in Wor1 activation between mother and daughter cells suggest the pace of Wor1 activation is determined early in the switching event. Once the expression of Wor1 in switching cells reached levels similar to stable opaque cells, they were always inherited across subsequent cell divisions indicating that the switch was complete.

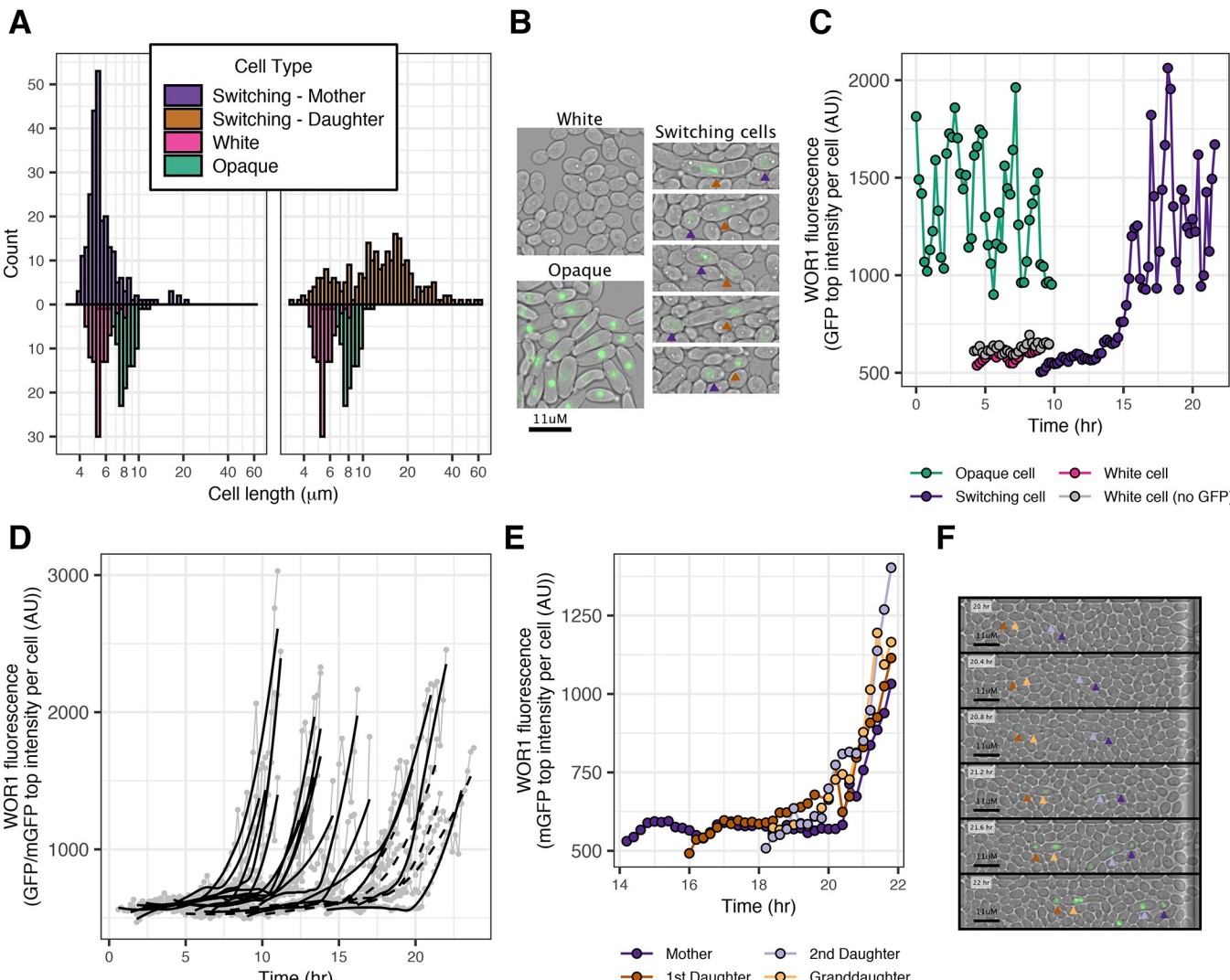

**Fig 2. Cell length and Wor1 fluorescence (GFP/mGFP) levels in single switching cells.** (A) Cell length distribution of switching cells compared to typical white and opaque cells. Switching initiates in pairs of mother and daughter cells, the left histogram shows the size distribution for mothers, the right, for daughters. Typical white and opaque cell distributions are shown beneath each histogram for reference. The strain contains the Wor1-GFP fusion protein. X-axis is logarithmic. (B) Representative images of white, opaque, and cells undergoing white to opaque switching. Arrows point out mother and daughter in each switching pair with colors corresponding to (A). Although elongated (pseudohyphal) cells are observed, they are neither necessary nor sufficient for switching. (C) Representative example of a Wor1 expression level trace in a single switching cell (purple). Traces of typical white (pink) and opaque (green) cells are shown for reference. Oscillations in high Wor1 levels occur with nuclear divisions. X-axis represents time from the beginning of the experiment; examples are taken from cells born at different times. The strain contains the Wor1-GFP fusion protein. (D) Examples of Wor1 activation in single switching cells. Traces (gray points and lines) are truncated to highlight the initial activation of Wor1. Black lines represent LOESS of individual traces. Examples are taken from 3 different experiments representing strains containing the Wor1-GFP (solid lines, *n* = 17) or the Wor1-mGFP (dashed lines, *n* = 4) fusion protein. (E) Representative example of a 4-cell (direct mother-daughter) pattern of synchronous Wor1 activation. X-axis represents time from the beginning of the experiment, and traces are shown starting at the birth of the mother cell. The strain contains the Wor1-mGFP fusion protein. (F) Subset of images representing data shown in (E) with time proceeding from top to bottom. Arrows point to individual cells with colors corresponding to (E). All 4 switching cells simultaneously activate Wor1. The data underlying this figure can be found at (https://osf.io/6e9vz/). GFP, green fluorescent protein; LOESS, locally estimated scatterplot smoothing.

## Certain lineages of cells show unusual switching patterns

An unanticipated feature of white-opaque switching emerged when we reconstructed complex pedigrees of switching cells, tracing the prior histories of cells destined to switch. In addition to the simple pairs of mother-daughter cells activating Wor1 simultaneously described above,

we observed instances where additional cells in the same trap also activated Wor1. Pedigree analysis revealed that these additional cells were closely related to the conventional mother-daughter switching pairs through a previous mother, daughter, or sister (Fig 3A). However, these related switching cells were separated by cell divisions that produced cells that did not switch, creating small mixed-fate pedigrees (Fig 3A).

To determine if these cases are the result of entirely independent switching events or are due to a complex pattern of switching within a single lineage, we analyzed the Wor1-mGFP strain, where overall switching rates are lower. We estimate the Poisson probability distribution for multiple switching events per trap based on the proportion of traps where there was no Wor1 activation (134/168 traps) (Fig 3B and see Methods). We find that the Poisson expectation (Fig 3B, white bars) of switching events is matched exactly when considering direct mother-daughter and mixed-fate pedigrees as single dependent switching events (chi-squared goodness of fit $p$-value, 0.986) (Fig 3B, light gray bars). However, the expectation is not matched if direct mother-daughter pedigrees are considered dependent but mixed-fate pedigrees are considered multiple independent events (chi-squared goodness of fit $p$-value, 0.0034) (Fig 3B, medium gray bars). This analysis rules out multiple independent events as the explanation for mixed-fate pedigrees and indicates that the switching events must be related to each other.

One common pattern of mixed-fate pedigree involved a mother cell budding 3 cells in succession (top-right "Mixed fate" pattern shown in Fig 3A). While the mother and the first and the third daughters ultimately activate Wor1, the second daughter does not (Fig 3C and 3D and S2 Video). With the Wor1-GFP fusion protein, we also observe similar patterns, including an example where a subset of the descendants of the second daughter also activates Wor1 (Fig F in S1 File). Considering all of the examples of mixed-fate pedigrees ($n = 7$) observed for the Wor1-mGFP strain (Fig G in S1 File), there were between 1 to 4 cell divisions separating switching cells from the establishment of the mixed-fate pedigree. These observations suggest that certain cells undergo a stochastic event that predisposes them and their direct descendants to a higher probability of switching then that observed in the general population. Whether or not cells in these pedigrees actually undergo switching appears to be determined by a second event. We do not know the basis of the first event (see Discussion), but in the next sections, we further explore the second step in which certain cells rapidly increase Wor1 expression.

## Using a synthetic inducible system in S. cerevisiae to investigate the properties of Wor1 autoregulation

Wor1 has been shown to bind to its own control region and to activate its own expression [31]. This positive feedback loop has been proposed to be a critical feature of white-opaque switching. Consistent with this hypothesis, we found that the frequency of white-opaque switching is reduced when a single Wor1 motif (out of 9) from the Wor1 control region is deleted (Table A in S1 File). In addition to Wor1, at least 7 other regulators bind the Wor1 control region [27,30], which complicates studying Wor1 autoregulation directly in *C. albicans*. To study Wor1 autoregulation without these confounding factors, we set up the Wor1 autoregulatory loop using a synthetic biology approach [38,39]. Specifically, we developed a system in *S. cerevisiae* where we can induce the expression of the *C. albicans* Wor1 protein with progesterone (Fig 4A and see Methods) and use flow cytometry to follow the activation of Wor1 transcription using a fluorescent reporter driven by the *C. albicans* Wor1 regulatory region.

The 7 kb control region (enhancer) of the *C. albicans* Wor1 was fused to the *S. cerevisiae* Cyc1 core promoter (see Methods) resulting in a reporter that was specifically activated by

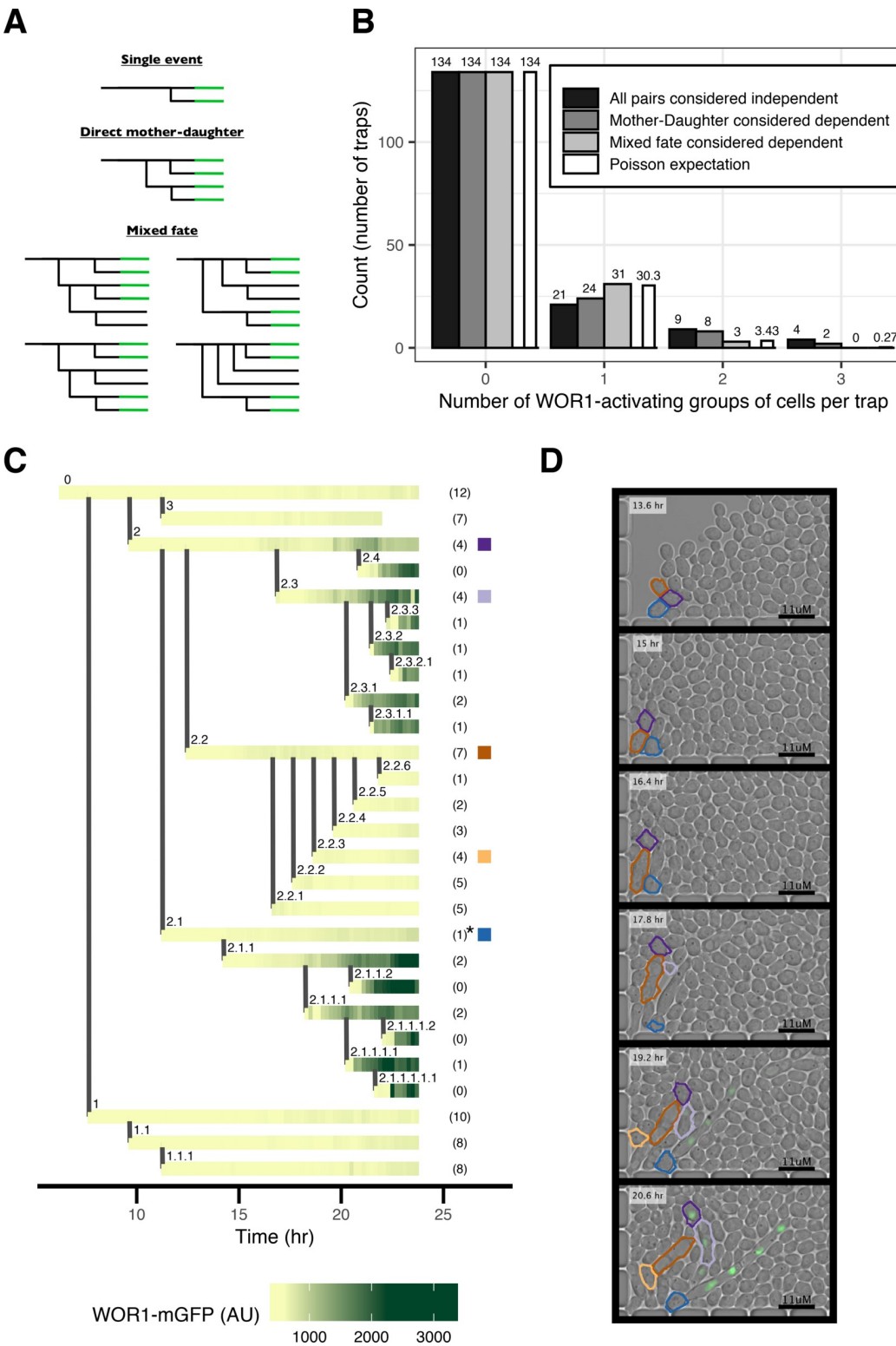

**Fig 3. Wor1 activation in pedigrees.** (A) Schematic of pedigree patterns of Wor1 activation, including several examples of mixed-fate pedigrees. Horizontal lines represent single cells; vertical lines represent budding of a daughter cell. Green color represents Wor1 activation. (B) Comparison of distributions of the number of switching events per trap to Poisson expectation (white bars). Different distributions are based on including different classes of pedigree patterns as single dependent events. The strain contains the Wor1-mGFP fusion protein. (C) Representative mixed-fate pedigree (top-right

"Mixed fate" pattern shown in (A)). Horizontal lines represent single cells; vertical lines represent budding of a daughter cell. Every horizontal line is made up of small tiles representing the measured Wor1-mGFP fluorescence at that time. Numbers within parentheses on the right of the pedigree represent the number of budded daughter cells per cell within the time period shown; not all daughters are depicted in the pedigree. The cell marked by the asterisk lost its nucleus to its daughter cell. Colored square tiles on the right-hand side indicate individual cells that are depicted in (D) using the same color scheme. (D) Subset of images representing data shown in (C) with time proceeding from top to bottom. Cell outlines are based on automated image analysis (see Methods). The data underlying this figure can be found at (https://osf.io/6e9vz/).

Wor1 expression (Fig H in S1 File). Using an mCherry fluorophore for the transcriptional reporter, we created independent (a or α mating type) strains where levels of Wor1, Wor1-GFP, Wor1-mGFP, or GFP could be controlled from a synthetic expression construct by adding hormone (see Methods). Comparing Wor1-GFP to Wor1-mGFP is informative as the ability of GFP to dimerize is the basis for increased switching in *C. albicans* with Wor1-GFP (Table A in S1 File). All GFP and mCherry measurements were normalized by forward scatter. As expected, GFP, Wor1-GFP, and Wor1-mGFP fluorescence all increased in a hormone-dependent manner when driven by the synthetic hormone-dependent construct (Fig 4B left and Fig I in S1 File). Despite identical transcriptional constructs, GFP was induced at 10 fold higher levels than Wor1-GFP and Wor1-mGFP (Fig 4B left), possibly due to differences in mRNA or protein stability or to translational efficiency. Inducing Wor1 or its GFP fusions also caused physiological effects such as cell clumping, leading to increased autofluorescence (in the Wor1 strains, orange lines in Fig 4B left) and bimodal induction distributions (in the Wor1-GFP strains, blue distributions in Fig I in S1 File). *S. cerevisiae* has 2 Wor1 homologs (*MIT1* and *ROF1*) with conserved DNA-binding domains and motifs. We created strains where both homologs are deleted and found that the results did not change significantly. We present data for both sets of strains.

## Wor1 autoregulation is ultrasensitive and the effect of GFP dimerization informs potential mechanism for increasing switching frequency

The Wor1 transcriptional reporter shows low levels of expression in the absence of hormone (Fig I in S1 File). As hormone levels increase, the reporter is activated in strains where the hormone induces Wor1 or its GFP fusions but not when GFP alone is produced (Fig 4B right). In strains inducing Wor1-GFP and Wor1-mGFP, we could measure both the level of the fusion protein (via the GFP channel) and the response of the Wor1 enhancer (via the mCherry channel) in single cells (Fig 4C and Fig J in S1 File). To quantify the response function of Wor1 autoregulation, we fit 4-parameter log-logistic functions (Hill equations) that describe S-shaped curves, to these distributions using least-squares estimation (Fig 4C and 4D and see Methods). The 4 parameters specify the lower horizontal asymptote, the upper horizontal asymptote, the value at which the response is half of the maximum, and the Hill coefficient, which is a measure of the steepness of the response curve (ultrasensitivity). Hill coefficients greater than 1 imply cooperativity in gene regulation. As reporter expression gradually decreases at high hormone levels in strains containing Wor1-GFP, we only used a subset of conditions for fitting in these strains (Fig J in S1 File). The decrease under these conditions suggests high levels of Wor1 dimers might negatively regulate Wor1 expression. All curves were ultrasensitive (Fig 4D) with mean Hill coefficient estimates of 3.2 (standard deviation, 0.4) for Wor1-mGFP and 4.3 (standard deviation, 0.9) for Wor1-GFP. While there was a small but significant increase in cooperativity for Wor1-GFP over that of Wor1-mGFP (*t* test *p*-value, 0.01), there was a striking difference in the level of protein required for half maximum activation, with 5 times higher levels required for Wor1-mGFP than Wor1-GFP (*t* test *p*-value,

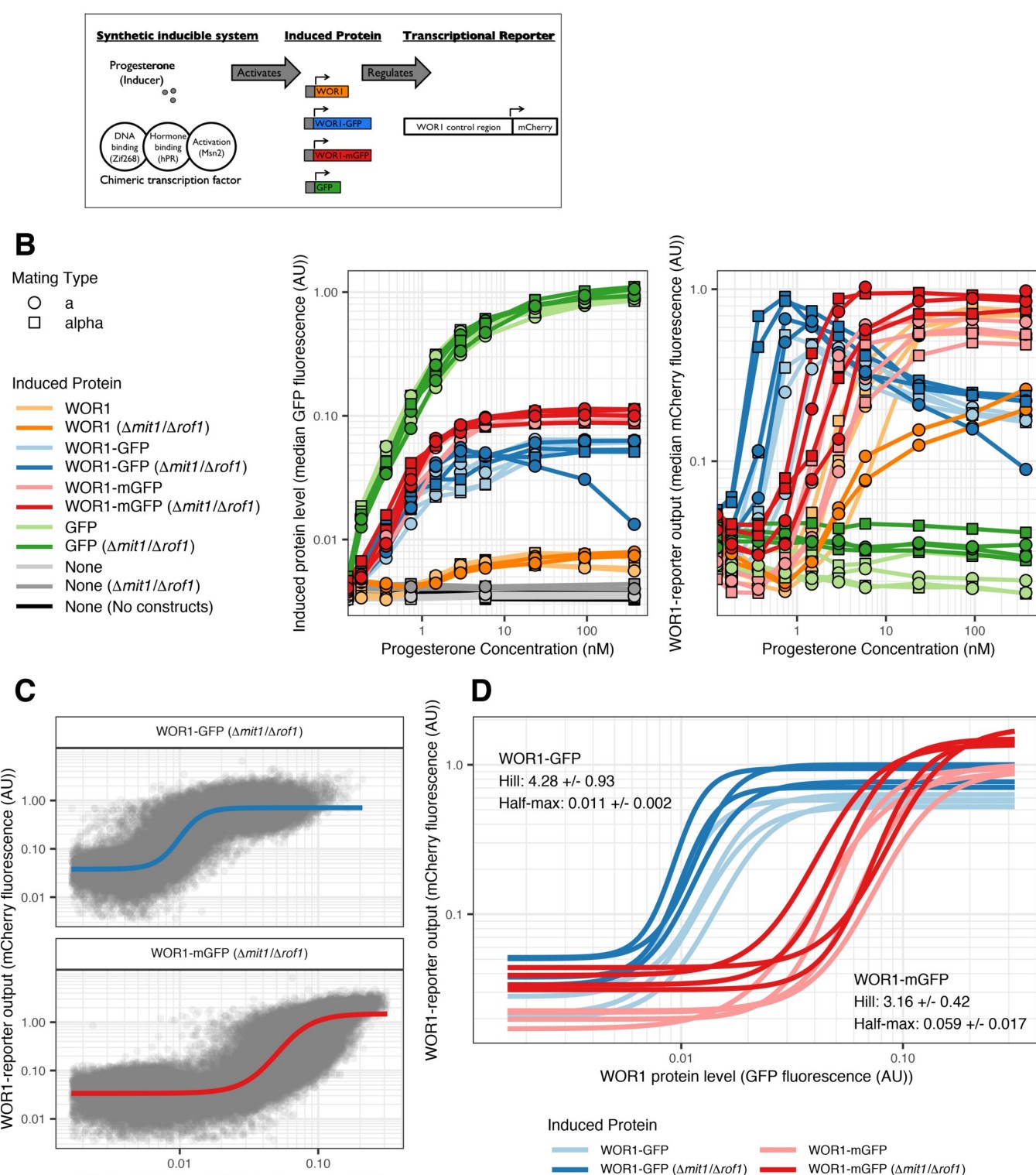

**Fig 4. Wor1 autoregulation parameters determined by a synthetic inducible system in *S. cerevisiae*.** (A) Schematic of experimental design. (B) Median GFP (left) or mCherry (right) fluorescence for strains inducing different proteins in a range of hormone concentrations. Every set of constructs is represented by 4 independently constructed strains, each measured on 2 different days (with the exception of the Wor1 (*Δmit1/Δrof1*) strains which were only measured once). X-axis and Y-axis are logarithmic. (C) GFP (Wor1 protein) and mCherry (Wor1 transcriptional activity) fluorescence for individual cells across hormone

concentrations. Data are shown for 2 strains inducing either Wor1-GFP (top) or Wor1-mGFP (bottom) measured on 1 day. Solid line represents fit Hill equation for each strain. X-axis and Y-axis are logarithmic. (D) Fit Hill equation curves for Wor1 transcription activity of strains inducing either Wor1-GFP (blue) or Wor1-mGFP (red). Each curve is fit to data measured on 1 of 2 different days. Inset text summarizes parameters for each set of strains (mean +/− standard deviation). X-axis and Y-axis are logarithmic. The data underlying this figure can be found at (https://osf.io/6e9vz/). GFP, green fluorescent protein.

$8.04 \times 10^{-5}$). As the ability of GFP to dimerize most likely increases the effective amount of Wor1 found in dimers, the difference in activation thresholds suggests that Wor1 normally acts as a multimer to efficiently bind DNA. Wor1-GFP levels in *C. albicans* opaque cells are comparable to the levels of Wor1-GFP needed to activate the reporter in our synthetic *S. cerevisiae* system (Fig J in S1 File), further indicating that the synthetic system captures features of the authentic *C. albicans* circuit.

## Discussion

The ability of cells to switch between different cellular states, each of which are faithfully inherited through many rounds of cell division, underlie a wide range of important phenomena including cell differentiation [40], antibiotic resistance [41,42], and cancer progression [43]. The fungal pathogen *C. albicans* and its close relatives *Candida dubliniensis* and *Candida tropicalis* all undergo a phenomenon known as white-opaque switching, whereby 2 distinct states are stable across many cell divisions, and switching between them is stochastic [44,45]. Although the presence of these heritable states has been preserved by selection in these species, the mechanism of switching and the complete range of physiological roles of the 2 cell types are topics of active investigation. It is known that opaque cells are the mating competent cell type [21] and are less susceptible to phagocytosis by macrophages, potentially promoting immune evasion [24,46]. Our recent work has also shown that, due to a specialized secreted protease program, a small number of opaque cells can promote growth of a large white cell population; thus, the 2 cell types can act synergistically [47]. Aside from its physiological roles, white-opaque switching has many advantages as a eukaryotic system to study stochastic phenotypic switching. It exemplifies 2 crucial phenomena, rare stochastic phenotypic switching and cell states that are stable through thousands of cell divisions. It shares many features with similar networks in multicellular eukaryotes, such as a large number of transcription factors and long, complex regulatory regions (enhancers). White-opaque regulators have also been shown to form phase-separated condensates [32], which are a feature of some mammalian enhancers [48]. A particular advantage of studying phenotypic switching and heritability in *C. albicans* is the relative ease of performing genetic manipulations. By combining genetic modifications with direct observation of single cells, our goal is to describe in quantitative terms the molecular mechanisms that underlie cell type switching.

Positive feedback, encompassing either positive interactions, double negative interactions, or autocatalysis is necessary, but not sufficient for the creation of heritable cell states [11]. To facilitate switching, the response must incorporate nonlinearity, also known as ultrasensitivity, in order to create sharp transitions from gradual changes [49]. Mechanisms for creating or increasing ultrasensitivity in regulatory responses include cooperative binding of transcriptional regulators [50–52], protein sequestration of regulators [53], and the number of steps in transcriptional cascades [54]. We show here that Wor1 regulation of its own promoter is ultrasensitive even in the absence of other *C. albicans* regulators. While our measurements provide a baseline for the autoregulation of Wor1, other factors in *C. albicans* may change these parameters. While difficult to rigorously test in the native *C. albicans* system, we propose this ultrasensitive response is a crucial feature of white-opaque switching that can be incorporated into mathematical models describing switching [55]. Furthermore, the propensity to dimerize

added by the GFP causes a small increase in the steepness of the activation curve but a large reduction in the activation threshold. It is this large reduction in activation threshold that likely leads to the large increase in white-opaque switching in *C. albicans* cells that harbor Wor1-GFP. This observation suggests switching depends on the equilibrium between Wor1 monomers and higher oligomers. While the natural threshold results in rare switching under standard lab conditions, our results show that switching frequency is easily adjusted by a relatively small change in the propensity of Wor1 to dimerize. Indeed, other white-opaque regulators may function in a similar manner to the GFP fusion; any increase in Wor1 multimer formation would produce a large effect on switching and could amplify a small amount of Wor1 expression. Although many questions remain concerning the molecular details of Wor1 autoregulation, our results quantify, for the first time, the ultrasensitivity of the WOR1 control region (enhancer) in response to carefully measured levels of the Wor1 protein.

A particular advantage of our study is the use of microfluidics, which has allowed us to observe white-opaque switching at a single-cell level and at high temporal resolution. The ability to track single cells and their lineages through time allowed us to make an unexpected observation that switching can occur in multiple groups of related cells, even when these groups are separated by cell divisions that produce cells that do not switch. The presence of this pattern did not differ between strains containing Wor1-GFP versus Wor1-mGFP despite different switching frequencies. We can draw 2 conclusions from this switching pattern. First, at least 2 stochastic events are needed for cells to switch from white to opaque. The first event defines small pedigrees of cells with increased probability of switching and the second event, which only occurs in a subset of cells in these pedigrees, directly leads to Wor1 activation and cell switching. Second, when a complete switching event occurs, several closely related cells in the pedigree switch simultaneously, a property that may be adaptive. The occurrence of coordinated switching would be advantageous in situations where the rapid generation of the rare cell type could be beneficial. Although our experiments do not address the adaptive value of white-opaque switching, increasing the number of opaque cells could be valuable both for immune evasion or metabolic cooperation with white cells. A similar switching pattern was observed in a lab-modified galactose utilization pathway in *S. cerevisiae* [56]. In that study, removal of negative feedback created a system dominated by a single positive feedback loop and led to cells stochastically turning on the GAL regulon in the absence of galactose. In a very similar manner to our observations, the authors saw correlated activation in cell pedigrees, at times separated by cells not activating GAL expression [56]. This comparison suggests these coordinated switching patterns might be a general feature in eukaryotic systems.

One hypothesis to account for our observed correlations among related cells relies on the stochastic and bursty nature of gene expression [57]. According to this hypothesis, the first stochastic event during white-opaque switching might be a rare burst of low-level Wor1 transcription. This state would be heritable over only a small number of cell divisions, defining the small pedigrees of cells predisposed to switch. The second stochastic event would then create variation in the amount of Wor1 between cells, perhaps because of variation in translation efficiency, uneven partitioning during division, or the stochastic expression of an inhibitor. Only certain cells in the pedigree would accumulate sufficient Wor1 protein to excite the Wor1 positive feedback loop, leading to high level Wor1 expression and subsequent cell type switching. This hypothesis is subject to explicit testing and will be the focus of future work.

Our work has resolved several issues needed for a quantitative molecular understanding of white-opaque switching and lays the foundation for subsequent hypothesis tests. While white-opaque switching and the genetic network that controls it are highly complex, it represents 1 of only a few cases of a true epigenetic change in a eukaryotic cell that does not require external

signals and is simple enough to be analyzed in detail and ultimately, understood in great depth.

## Materials and methods

### *C. albicans* strain construction

The genetic background for *C. albicans* is SC5314. SNY425 was used as a prototrophic a/α reference strain [58]. For a prototrophic a/Δ reference strain, the α copy of the MTL in SNY250 [59] was replaced with a copy of *ARG4* using pJD1 [60]. Genetic modifications were created using a *SAT1* marker-based CRISPR protocol targeting *Candida maltosa LEU2* [61]. Homozygous Wor1-GFP and Wor1-mGFP fusion proteins were created at the endogenous *WOR1* locus using a combination of 2 guides encompassing the stop codon (`GGTACTTAGTTGAATT AATA` and `GTACTTAGTTGAATTAATAC`). Donor DNA was created by restricting plasmids containing the GFP (or mGFP) sequence flanked by 500 bp of homology to the 3′ end of the *WOR1* ORF and 3′ UTR [62]. The *WOR1* motif deleted from the *WOR1* promoter consists of 3 overlapping binding motifs (17 bp, `AAAGTTTAAACTTTAAA`), found 5,802 bp upstream of the translation start site. A combination of 2 guides were used for creating the deletion (`GAAATTTATGAAAGACGGGT` and `ACCAGTAGTGATTCATAAAT`), donor DNA consisted of 90 bp oligos containing 45 bp homology to flanking sequences with 2 single bp substitutions to disrupt PAM or guide sequences (`AATAATTAGAGTTTTACAAGAAATTTATGAAAGACG GGTGcGAAATACTATAAAATAGAGCAATAAATGACAGAACCTAgCAGTAGTGAT`).

### *S. cerevisiae* strain construction

The genetic background for *S. cerevisiae* experiments is S288C. Starting strains were mating type a or α with multiple auxotrophies, created by mating FY23 (*leu2Δ1 ura3-52 trp1Δ63*) [63] and a mating type α spore of BY4743 (*his3Δ1 leu2Δ0 lys2Δ0 met15Δ0 ura3Δ0*) [64]. Subsequent transformations integrated various constructs in the following order: the chimeric transcription factor, deletion of the *WOR1* homologs, the induced protein (GFP, Wor1-GFP, Wor1-mGFP, or Wor1), and the *WOR1* transcriptional reporter. Plasmids were constructed using either restriction digest and subsequent ligation or by using homology-based cloning (In-Fusion, Takara 638911) and were sequence verified, all PCRs utilized a high fidelity polymerase (CloneAmp HiFi PCR Premix, Takara 639298). Plasmids contained components of the synthetic inducible system flanked by homology to the intended genomic integration locus and were restricted prior to transformation. The chimeric transcription factor used is "Z3PM" [39] that consists of the Zif268 DNA-binding domain, hPR ligand-binding domain and the yeast Msn2 activation domain. The transcription factor was expressed from the *S. cerevisiae ADH1* promoter and terminated by the *C. albicans ADH1* terminator. The transcription factor was integrated into *leu2* proceeded by the *Candida glabrata LEU2* sequence, including promoter and terminator. Induced proteins were expressed from a modified *GAL1* promoter containing 3 dimeric Zif268 binding sites [39] and terminated by the *C. albicans ADH1* terminator. The proteins were integrated into the *trp1* locus proceeded by the *C. glabrata TRP1* sequence, including promoter and terminator. The *WOR1* homologs (*MIT1* and *ROF1*) were knocked out using a plasmid-based CRISPR approach [65]. A plasmid with a 2 μ origin of replication was constructed that contained Cas9, a KanMX selection cassette and 2 guides (1 for each of the homologs). The guides were expressed from the *SNR52* and *SUP4* promoters, respectively, and both were terminated by a copy of the *SNR52* terminator. Donor DNA consisted of 90 bp complementary annealed oligos [61] containing 44 bp of homology upstream and downstream of the ORF, flanking a GG sequence. After verifying both homologs were deleted, cells were grown overnight in the absence of G418, plated on YPD and replica plated

to YPD+G418 to identify colonies that had lost the *CAS9* containing plasmid. The main *WOR1* transcriptional reporter consists of 6656 bp of the *WOR1* promoter (ending at a presumed TATA box, 2048 bp before the translational start site), 115 bp containing the end of the *CYC1* promoter and it is 5′ UTR (starting at the "−52" TATA box [66]) and a fluorescent protein ORF (mCherry or GFP) followed by the *C. albicans* *ACT1* terminator. An alternative reporter contained an additional 70 bp of the end of the *WOR1* promoter and 1,980 bp encompassing the *WOR1* 5′ UTR instead of the *CYC1* sequence. Transcriptional reporters were preceded by a hygromycin resistance cassette and integrated into the *ura3* locus. Integration was achieved using 2 separate restricted plasmids simultaneously transformed. One plasmid contained homology upstream to the *URA3* locus, the hygromycin cassette and the *WOR1* promoter and beginning of 5′ UTR. The second plasmid contained the end of the *WOR1* promoter, the *CYC1* (or *WOR1*) 5′ UTR, the fluorescent protein and terminator and homology downstream to the *URA3* locus. With 1 exception, combinations of constructs were represented by 2 completely independent strains based on different starting strains (of mating type a or α). The induced Wor1 (Δ*mit1*/Δ*rof1*) combination was represented by 2 mating type a strains.

## Microscopy and microfluidics

*C. albicans* strains were streaked from frozen stocks on SD+aa+uri plates and incubated for 5 to 9 days at 25˚C. Colonies were directly diluted to an OD of 0.25 and grown in liquid SD+aa +uri for 4 hours. Cultures were washed and diluted to approximately $5 \times 10^6$ cells/ml in PBS without calcium or magnesium, and 50 ul were loaded into microfluidic plates. For microfluidic experiments, we used a CellASIC ONIX microfluidic platform and corresponding Y04T-04 plates. Each plate contains 4 separate chambers, each chamber containing 104 individual traps. Images were captured using a Nikon Ti2-E microscope equipped with a Photometrics Prime 95B-25mm Camera and a CFI60 Plan Apochromat Lambda 100× Oil immersion objective lens. Cells were loaded into the plate chambers using 55.1 kPa of pressure for 5 seconds, and media (SD+aa+uri) flow was started immediately. The media flow program consisted of cycles of three 5-second bursts of perfusion at 10 kPa from each of 6 inlet channels followed by an hour of perfusion at 1.7 kPa from each of 6 inlet channels. Coordinates for fields containing single traps loaded with at least 1 cell were manually determined. Nikon NIS Elements software was used to drive stage movement and acquisition; the Nikon Perfect Focus System was utilized. Images (DIC and GFP channels) were captured for each field (40 to 70 per chamber) every 12 minutes for 24 hours. Temperature was kept at 25˚C using an OKOLab Cage Incubator.

## Image and data analysis

Custom Matlab scripts were used for image analysis; statistical analysis and data visualization utilized R [67], with extensive use of *tidyverse* [68]. Images were first adjusted to correct for stage movement. Each image was compared to the previous image in the time series, and the optimal geometric transformation was found to account for any (x,y) movement (Matlab function *imregtform*). Next, cells were segmented automatically in each image. Cell borders were identified using a combination of bottom hat filtering, top hat filtering, dilation, erosion, and selection based on object properties (area, perimeter, and eccentricity). Individual cells were tracked through time by manually determining their position in a series of images; each position was then associated with the corresponding automatically segmented cell. Pedigree information was also manually determined. For a metric of Wor1 expression per cell, the 300 highest intensity pixels (in the GFP channel) were averaged. Typical white cell size is 1,750

pixels; typical opaque cell size is 2,500 pixels. Cell length (Fig 2 and Fig B in S1 File) was determined using Fiji [69] by recording the length of a line drawn along the cells long axis. Cell length was recorded when cells were beginning to bud a new cell. A total of 100 references each of white and opaque cells were recorded. Cell length was recorded for all pairs of switching cells across 3 experiments from different days, resulting in 248 mother-daughter pairs of cells containing Wor1-GFP and 50 pairs of cells containing Wor1-mGFP. For determining if mixed-fate pedigrees represent multiple independent switching events, we considered 168 traps loaded with cells containing Wor1-mGFP. The numbers of switching events were counted in each trap according to 3 different ways of calculating independent events: (1) every pair of mother-daughter cells are considered independent; (2) pairs of direct mother-daughter pairs are no longer considered independent; and (3) pairs in mixed-fate pedigrees are also no longer considered independent. We calculated the expected Poisson probability distribution based on the unambiguous zero-class of 134 traps ($\lambda = -\log (134/168)$). We performed chi-squared goodness-of-fit tests with $p$-values computed by Monte Carlo simulations with 2,000 replicates.

## Flow cytometry, experiments, and data analysis

*S. cerevisiae* strains were streaked from frozen stocks on YPD plates and incubated for 2 days at 30˚C. Colonies were inoculated into 3 ml of SD+aa+uri media and grown overnight. Cells were washed, resuspended in PBS without calcium or magnesium, and diluted 1:16 into 96-well plates containing SD+aa+uri with different concentrations of progesterone. Progesterone was diluted from 2 mM stocks (95% ethanol, 5% DMSO); lower concentrations were supplemented with ethanol and DMSO to match the high concentration. Plates contained 8 experimental strains measured at 10 different concentrations of progesterone and 3 control strains measured at 4 concentrations of progesterone. For each set of strains, 2 identical plates were created and incubated, shaking at 500 rpm at 30˚C. After a set incubation time, plates were washed 3 times and diluted in PBS without calcium or magnesium with 1 mM EDTA. One plate was read after 4.5 hours of incubation, and the second was read at 24 hours. Data from both plates were similar, and the 24-hour readings are presented. Measurements were taken using a BD FACSCelesta. GFP fluorescence was measured by blue (488 nm) excitation and a 530/30 band pass emission filter. mCherry fluorescence was measured by yellow-green (561 nm) excitation and a 610/20 band pass emission filter. Flow cytometry data were analyzed using R [67], with extensive use of *tidyverse* [68]. Data were imported into R using *flowCore* [70]. Data were filtered to remove potential cell debris, cell aggregates, and contaminants by constructing 2 gates, 1 based on forward and side scatter and a second based on GFP and a violet (405 nm) excitation, 670/30 emission channel. GFP and mCherry measurements were normalized by forward scatter. Presented data encompass 376 wells with a range of 127 to 8,905 events per well after filtering (median: 7,095.5; 95% of samples have over 4,000 events). For fitting Hill equations, we used the *drm* function in the package *drc* [71] using robust median estimation. mCherry measurements were log10 transformed and fitted to a 4-parameter log-logistic function with a parameterization converting the natural log of the half-max value into a parameter. For Wor1-GFP, only data from progesterone concentrations less than 2 nM were used for fitting.

## Supporting information

**S1 File.** A PDF file containing Supporting information figures (A–J) and table (A). Each figure is followed by a legend and data underlying the figures can be found at (https://osf.io/6e9vz/). Table A. Switching frequencies. Switching frequencies as calculated by counting sectors on

colonies grown on solid agar plates. Data come from 3 independent experiments containing multiple plates per strain. Numbers of colonies and sectors are pooled across plates. The percent of colonies with at least 1 sector is calculated. Standard error is the standard error of a sample proportion ($\sqrt{[p(1-p)/n]}$). Fig A. Opaque cell growth in microfluidic traps. Two examples of microfluidic traps with growing opaque cells. Strain contains Wor1-GFP fusion protein. Fig B. Cell length distributions. (A) Cell lengths of pairs of switching cells (mothers versus their daughters), including switching cells containing Wor1-mGFP. Typical white and opaque cell distributions are shown on margins. (B) Cell length distributions (violin plots) of different cell types. For the Wor1-GFP strain, data come from 3 independent experiments that were pooled in main text Fig 2A. Fig C. Nucleus loss during mitosis in switching cells. (A) Representative example of a cell division where the mother cell nucleus enters the elongated daughter cell but returns to the mother cell. Strain contains Wor1-GFP fusion protein. (B) Representative example of cell divisions where the mother cell nucleus enters the elongated daughter cell and remains in the daughter cell, creating a polyploid cell. There are 2 cell divisions with this pattern in this example. Strain contains Wor1-GFP fusion protein. Fig D. Semi-automated image analysis pipeline. (A) Representative original images of 2 time points for 1 field. (B) Same images as (A) after adjustment for (x,y) movement (note black regions at image edge) and automated segmentation of cells (magenta colored lines defining cell borders). (C) Same images as (A) and (B) with overlay of manually defined cell identities, pedigree information is explicitly encoded in cell names (e.g., cell "2.1" is the first daughter of cell "2," which, in turn, is the second daughter of cell "0"). Fig E. Fluorescence (GFP/mGFP) levels in single cells. (A) Additional Wor1 expression level traces in single cells. Traces are of white cells that do not contain Wor1-GFP (gray) and white cells that do contain Wor1-GFP (pink/purple). X-axis represents time from cell birth. Cells that are not actively switching have similar fluorescence levels to typical white cells that do not contain GFP. (B) The same examples of Wor1 activation in single switching cells shown in main text Fig 2D, lined up by the approximate time cells reach high Wor1 levels. Although the time between the birth of a switching cell and its switching varies, reaching high levels of Wor1 from background fluorescence levels takes approximately 3 hours. Fig F. Mixed-fate pedigree example, Wor1-GFP. (A) Representative pedigree in which multiple groups of cells activated Wor1. Horizontal lines represent single cells; vertical lines represent budding of a daughter cell. Every horizontal line is made up of small tiles colored by Wor1-GFP fluorescence. Numbers within parentheses on the right of the pedigree represent the number of budded daughter cells per cell within the time period shown; not all daughters are depicted in the pedigree. An asterisk represents that the cell lost its nucleus to its daughter cell. Colored square tiles single out particular cells that are depicted in (B). (B) Subset of images representing data shown in (A). Particular cells are outlined in colors corresponding to (A). Cell outlines are based on automated image analysis. Fig G. Mixed-fate pedigrees. Schematics of all mixed-fate pedigrees observed for the Wor1-mGFP strain. Numbers represent the number of cell divisions separating switching cells from the establishment of the predisposed pedigree. Representation is for illustrating cell relationships and do not accurately reflect cell division times or timing/intensity of Wor1 activation. Fig H. Alternative Wor1 transcriptional reporters in *S. cerevisiae*. Distributions of Wor1 transcriptional reporters; all strains shown are inducing Wor1 protein with increasing progesterone concentrations and were independently constructed from the strains shown in the main text. "Full" reporter contains both the Wor1 7 kb control region and promoter and the Wor1 2 kb 5′ UTR. The "No-UTR" reporter contains the Wor1 control region and the Cyc1 core promoter and 5′ UTR as explained in the Methods section. Fig I. Distributions of Wor1 protein and reporter levels. Distributions of GFP (A) and mCherry (B) fluorescence; data are shown for all strains in a subset of 4 hormone concentrations (nM progesterone). Note bimodal distributions of GFP fluorescence when inducing

Wor1-GFP at high hormone concentrations in (A). Note low-level expression of the Wor1 transcriptional reporter compared to autofluorescence in (B). Fig J. Distributions of Wor1 protein and reporter levels. (A) Same data as main text Fig 4C, separated by hormone concentration (nM progesterone). (B) 2D density plots of flow cytometry data. *S. cerevisiae* Wor1-GFP and Wor1-mGFP data (same as A) are shown alongside data for white and opaque *C. albicans* cells containing a Wor1-GFP fusion protein and a transcriptional reporter encompassing the 7 kb control region of Wor1 followed by mCherry (no 5′ UTR sequence).
(PDF)

**S1 Video. Example of time-lapse movie of a switching event Movie corresponds to "Example 1" in Fig 1D.** Movie is shown twice; in the second repetition, tracked cells are labeled. This strain contains the Wor1-GFP fusion protein.
(MP4)

**S2 Video. Example of time-lapse movie of a switching event containing a mixed-fate pedigree Movie corresponds to example in Fig 3C.** Movie is shown twice; in the second repetition, tracked cells are labeled. This strain contains the Wor1-mGFP fusion protein.
(MP4)

## Acknowledgments

We thank members of the Johnson lab, in particular, Matt Lohse and Kyle Fowler for helpful discussions and Ananda Mendoza for technical support. We thank David Gresham, Kerry Geiler-Samerotte, Mariana Gomez Schiavon, and Ranen Aviner for advice and the lab of Hana El-Samad and Dmitri Petrov for providing reagents.

## Author Contributions

**Conceptualization:** Naomi Ziv, Alexander Johnson.

**Formal analysis:** Naomi Ziv.

**Funding acquisition:** Alexander Johnson.

**Investigation:** Naomi Ziv, Lucas R. Brenes.

**Supervision:** Alexander Johnson.

**Visualization:** Naomi Ziv.

**Writing – original draft:** Naomi Ziv.

**Writing – review & editing:** Naomi Ziv, Lucas R. Brenes, Alexander Johnson.

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
