## [Editor Report · Decision Letter 0]

16 Dec 2021

Dear Naomi, 

Thank you for submitting your manuscript entitled "Multiple molecular events underlie stochastic switching between two heritable cell states in a eukaryotic system" for consideration as a Research Article by PLOS Biology.

Your manuscript has now been evaluated by the PLOS Biology editorial staff, as well as by an academic editor with relevant expertise, and I'm writing to let you know that we would like to send your submission out for external peer review.

IMPORTANT: We think that because of its somewhat preliminary (but tantalising) nature, it would be better to review this paper as a Discovery Report. No re-formatting is required at this stage, but please could you change the article type to "Discovery Report" when you upload the additional metadata (see next paragraph)?

Once your full submission is complete, your paper will undergo a series of checks in preparation for peer review. Once your manuscript has passed the checks it will be sent out for review. To provide the metadata for your submission, please Login to Editorial Manager (https://www.editorialmanager.com/pbiology) within two working days, i.e. by Dec 20 2021 11:59PM.

If your manuscript has been previously reviewed at another journal, PLOS Biology is willing to work with those reviews in order to avoid re-starting the process. Submission of the previous reviews is entirely optional and our ability to use them effectively will depend on the willingness of the previous journal to confirm the content of the reports and share the reviewer identities. Please note that we reserve the right to invite additional reviewers if we consider that additional/independent reviewers are needed, although we aim to avoid this as far as possible. In our experience, working with previous reviews does save time. 

If you would like to send previous reviewer reports to us, please email me at rroberts@plos.org to let me know, including the name of the previous journal and the manuscript ID the study was given, as well as attaching a point-by-point response to reviewers that details how you have or plan to address the reviewers' concerns. 

Given the disruptions resulting from the ongoing COVID-19 pandemic, please expect some delays in the editorial process. We apologise in advance for any inconvenience caused and will do our best to minimize impact as far as possible.

Kind regards,

Roli

Roland Roberts

Senior Editor

PLOS Biology

rroberts@plos.org

---

## [Decision Letter · Decision Letter 1]

17 Feb 2022

Dear Dr Ziv,

Thank you for submitting your manuscript "Multiple molecular events underlie stochastic switching between two heritable cell states in a eukaryotic system" for consideration as a Discovery Report at PLOS Biology. Your manuscript has been evaluated by the PLOS Biology editors, an Academic Editor with relevant expertise, and by two independent reviewers.

You'll see that both reviewers are broadly positive about your study, but each suggests a number of minor additional experiments and analyses that will be needed for further consideration. We also invited the reviewers to cross-comment on each other's reviews; reviewer #2 sent some additional comments that we have included, as you might find them helpful to improve your paper.

In light of the reviews (below), we are pleased to offer you the opportunity to address the comments from the reviewers in a revised version that we anticipate should not take you very long. We will then assess your revised manuscript and your response to the reviewers' comments and we may consult the reviewers again.

IMPORTANT: We considered your paper as a Discovery Report. The maximum allowable number of Figs for this article type is 4, but you currently have 5. Please either combine some of these Figs or move some material to the supplement.

We expect to receive your revised manuscript within 2 months.

**IMPORTANT - SUBMITTING YOUR REVISION**

*Resubmission Checklist*

*Published Peer Review*

*PLOS Data Policy*

*Blot and Gel Data Policy*

Sincerely,

Roli Roberts

Roland Roberts

Senior Editor

PLOS Biology

rroberts@plos.org

REVIEWERS' COMMENTS:

Reviewer #1:

[see attachment for fully formatted version]

Stochastic fate switching is the basis for the formation of non-genetic phenotypic heterogeneity. While multiple instances of such phenotypic switching are observed across several model systems, the molecular basis still remains elusive. In this paper, Ziv et al used the white to opaque switching phenotype in Candida albicans to study the mechanism that leads to such heterogeneity by focusing on the inheritance and positive autoregulation of Wor1 expression across generations. High expression of Wor1 is associated with the formation of opaque cells. The paper reports and tries to link to distinct phenomena: the pattern of switching in Candida cells and the ability of the product of a transplanted Wor1 gene to activate its own transcription in Saccharomyces cerevisiae. The cerevisiae experiment reveals very strong positive feedback and it is possible that both the same positive feedback exists in Candida but given the large phylogenetic distance between the organisms this is a hypothesis more than either an inference or a conclusion and the writing of the paper should clearly reflect this distinction even though this is a Discovery article.

The paper reports that there appears to be an event that occurs before the detectable increase in Wor1 levels and which increases the probability that Wor1 will rise and cells will become opaque. The most interesting observation is that in a lineage where this event has occurred, some cells can activate Wor1 and others cannot but that all the cells that activate Wor1 do so synchronously. The authors argue that there must be two events to describe this behavior, but these could be closely related. For example, the first event could be inducing Wor1 expression to a level below their threshold of detection that leads to a slow, but essentially deterministic increase in Wor1 concentration that will eventually lead to strong Wor1 expression and a commitment to the opaque state. To explain the failure of some cells in a lineage to make this switch, you need only appeal to the stochastic expression of some inhibitor that interrupts this positive feedback loop.

There are two principal weaknesses to the paper. The first is whether the key event that leads to opaque cells exceeding a certain level of Wor1 expression, something else, or Wor1 expression plus something else. Testing necessity would be hard but the observation that Wor1 dimerization, through GFP, increases switching supports the notion that increased Wor1 expression stimulates switching. The second is the large gap between the Candida and Saccharomyces experiments. The behavior in Saccharomyces is suggestive but not much more and it’s unclear how much measuring a Hill coefficient in a different organisms amplifies previous work that demonstrated positive autoregulation of Wor1 expression. Whether these weaknesses preclude publication as a Discovery article is an editorial decision.

Major recommendations:

1. An interesting question would be to see if the current data can put a limit on how many generations the state that leads to later synchronous Wor1 activation can exist for. To put it the other way, how distantly related can two cells in a lineage be that both participate in synchronous Wor1 activation. 

2. A convenient way to confirm bistability arising due to the positive autoregulation of Wor1 in Saccharomyces would be to use the flow cytometry data used for Figure 5B. If one induces Wor1-GFP to a level that is around the half-maximal threshold, the expression level of mCherry driven from the Wor1 promoter at later generations should be heterogeneous i.e. some cells in the population should show strong mCherry expression and other should have none. An important control would be to show all cells have mCherry OFF in low Wor1-GFP expressed strain and all cells have mCherry ON in high Wor1-GFP expressed strain. This is an easy experiment to do with the strains reported in the paper

Minor comments:

1. In Figure 1D, the authors can put Wor1-GFP in one panel and Wor1-mGFP in the other rather than two examples to show the differences in switching rate between the two constructs. 

2. Line 235, Page 215: Is the formation of polyploid cells associated with pseudohyphae formation. It might be interesting to some readers if the authors observe something similar. 

3. Line 264, Page 216: The authors have mentioned here ‘they were always inherited across subsequent cell divisions’ but later they show that some of the cells in the lineage don’t turn on Wor1. The authors should revise this statement to avoid inconsistency. 

4. Line 252, Page 216: It is not exactly clear from the figure 3A-C if the time taken to each maximum level for Wor1 is 3 hours. In Figure 3C, the mother cells take ~6 hours to reach maximum expression levels. Therefore, it would be optimal to put a distribution of switching time and present the mean/standard deviation. 

5. In the figure 3B, the data can be plotted starting from the cell division rather than imaging time to show that they more or less take the same amount of time to reach activation thresholds. The mean and standard deviation of the time can be put on the graph. Additionally, the plots corresponding to mGFP and GFP need to be colored separately. 

Reviewer #2:

In this manuscript, Ziv et al. investigate the stochasticity of Wor1 activation in Candida albicans which leads to rare switching from the white to opaque cell state. Wor1 is a master regulator of the opaque cell state. This is a Discovery Report and so the authors don't claim to have fully worked out the molecular basis of this stochasticity but instead report progress on this question by a combination of a few new techniques. They argue that this environmental-induction free system is a useful model for understanding the basis of stochastic switching and the consequences of epigenetic changes to cell state.

The authors used single-cell microscopy with GFP labeled Wor1 to monitor Wor1 activation. Tracking cells (and their progeny) over time was a key part of their approach and so they used a microfluidic device with replicated cell growth chambers to enable them to follow individual cells over time while also having large enough numbers to be able to find rare stochastic switching events. They find that dimerizing GFP increases the switching rate, and so they use this to increase the number of events they can observe while also validating their findings using a non-dimerizing GFP variant. They find that not only does Wor1 increase in expression in a subset of cells (as expected) but also that this state is inherited over two generations (they report data from a single granddaughter and two daughters). Often, seemingly independent switching lineages in the same cell trap were actually related a few more generations back. However, not all lineages descended from that previous generation switched, so while there is likely some protein or cell state that is inherited (supported by their Poisson analysis), it clearly does not surpass some threshold such that all descendants would be activated. It isn't clear what this cell state is yet.

The authors construct a synthetic Wor1 positive autoregulatory reporter in S. cerevisiae and use it to estimate the Hill coefficient and the level of protein at half-max activation of the Wor1 promoter by Wor1. They find a big different in protein at half-max activation between the GFP-dimerized Wor1 and the non-dimerizing version.

They propose a two-step model for Wor1 activation with a potentiating phase where the cell state of some cells changes. These cells and their progeny are predisposed to switch, but only some end up doing so after another molecular event.

Overall, this seems a fairly clean paper. Microfluidics is a sensible technique to capture and track these events, particularly since they want to look at lineages of cells. And their use of S. cerevisiae to study whether cooperativity might be involved is also reasonable.

The pseudohyphal growth and mitosis part of the paper (204-240) felt much less satisfying than the rest of the paper. On the one hand, the association of pseudo-hyphae and GFP in the videos is unmistakable and so the authors obviously need to discuss it. They claim (and show to some extent in figure 2A and also point to their videos) that pseudo-hyphae are neither necessary nor sufficient for switching. It might be nice to given some percentages or a 2x2 table with the data on this. They also point out some unusual nuclear movement between mother and daughter during division. Because this Discovery Report (as far as I understand) is partly meant to spur new research, it seems appropriate to leave this story unresolved. But it clearly merits follow-up - particularly as the pseudohyphal elongation seems to be transient. A minor point on figure 2A - I wonder if a scatterplot of mother-daughter size might be more informative than separating them like they are. The white/opaque histograms could still be on the margins.

On the basis of their cell traces over lineages, the authors claim that mothers and daughters increase Wor1 expression simultaneously (260). Fig 3C is meant to show this. But it seems to me that there is variation on the order of a few hours between these. Is the order consistent across different sets of 4 cells? Is it related to size of the cells or concentration of GFP? They say that this supports the idea that the rate of switching is set early and continues apace even after cell division events (264). I wonder if there is anything more that can be done with this data to try to pin down when this second stochastic event happens relative to the first division of the mother. 

In the discussion, on line 511 they propose that several cells switching at once may be adaptive. This seems premature. It may be adaptive or it may not be. There are a few thresholds in this model and some fractions of switching cells in a pedigree against a background of non-switchers and none of the evidence presented suggests that these have been fine tuned by selection.

Lines 524-535 present an interesting hypothesis tying the second stochastic step to translation efficiency and then points out that the Wor1 UTR reduces protein expression. I'd recommend putting this fact at the start of this paragraph since it is important context for the model that lines 524-533 lay out.

All in all, this paper describes a logical series of experiments that seems to fit the Discovery Report brief. I have the suspicion that they can tease more out of their movies about the pseudohyphal connection, but that isn't essential and I may be wrong. 

Minor point - figure 2C, the first measurements for the two daughters have very low GFP. Is this real or is it some artifact of the image processing? If taken at face value, It suggests that GFP is very unevenly distributed between mother and daughter but the daughter GFP recovers one image later to be the same as the mother's.

REVIEWER #2'S CROSS-COMMENTS:

Thanks for the opportunity to see the other review and comment. I think reviewer 1 caught several things that I missed. Their comments bring a few more ideas to mind.

Reviewer 1 mentions a scenario that involves Wor1 expression below a threshold for detection. I wonder if the authors could determine what this threshold would be. This made me go back to figure 3. There is an odd mismatch between the GFP levels of the no-GFP cells in panel A and the trajectories in panels B&C which appear to start consistently rising below this no-GFP level. It might be useful to show more trajectories like figure C, for both cells that will switch and cells that will not (like the grey circles in A). One difference between A and C is GFP vs. mGFP and so it would be good to know if the detection threshold somehow depends on which GFP variant.

This question of detection thresholds also brings up an anomaly in Figure 5B. Why are the orange lines above the grey ones? Orange shouldn't have GFP.

Reviewer 1 notes the large phylogenetic distance between C. albicans and S. cerevisiae in the engineering experiments. The argument is that the C. albicans transcription factor will act on the C. albicans regulatory sequence differently in S. cerevisiae than in C. albicans in such a way as to make the Hill coefficient estimates suspect. I don't think

phylogenetic distance per se is a problem. The question is whether there are other factors in C. albicans (or in S. cerevisiae) that are hindering (or helping) Wor1 activate its promoter. The control suggests that there aren't other factors in S. cerevisiae that are activating it on its own. So any effect would have to come through interaction with the Wor1 protein itself. But I think identifying any such factors if they exist would be beyond the scope of this paper. I think this experiment and the conclusions they draw are fine and probably better than if they had tried a similar sort of thing (e.g. measuring binding) using in vitro techniques. But the more careful phrasing and explanation of the inference and more explanation of how this is an advance over previous knowledge that Reviewer 1 asks for is warranted.

Reviewer 1 presents a model for the two events that involves first a triggering event in a subset of cells that leads to increase in Wor1 levels and then a random failure of inhibition at some point right around the threshold that would let Wor1 activate itself. The authors didn't commit to any particular mechanism for these two events, and so I think they could add this possibility in their discussion. I wonder if candidates for this model could be identified - it would probably have to be a constitutive regulator where knockouts lead to increased opaque switching. There is probably data out there to narrow down to possible candidates, if any. I suppose it could also be induced in the first event along with Wor1. But all this is speculation.

---

## [Editor Report · Decision Letter 2]

25 Apr 2022

Dear Dr Ziv,

Thank you for submitting your revised Discovery Report entitled "Multiple molecular events underlie stochastic switching between two heritable cell states in a eukaryotic system" for publication in PLOS Biology. Your revisions and responses have been assessed by me and the Academic Editor. 

Based on this assessment, we will probably accept this manuscript for publication, provided you satisfactorily address the remaining points raised by the reviewers. Please also make sure to address the following data and other policy-related requests.

a) Given that your work is conducted in Candida and Saccharomyces, we suggest that you change your title to "Multiple molecular events underlie stochastic switching between two heritable cell states in fungi."

b) Please address my Data Policy requests below; specifically, we need you to supply the numerical values underlying Figs 2ACDE, 3BC, 4BCD, S2AB, S5AB, S6A, S8, S9, S10AB; it’s unclear whether these are available in your OSF deposition, which seems to mostly comprise raw data and scripts. Please clarify and/or supply these values, either as a supplementary data file or as part of the OSF package.

c) Please also cite the location of the data clearly in each relevant main and supplementary Fig legend, e.g. “The data underlying this Figure can be found in https://osf.io/6e9vz/” or “The data underlying this Figure can be found in S1 Data.” 

We expect to receive your revised manuscript within two weeks. 

*Published Peer Review History*

*Press*

Sincerely,

Roli Roberts

Senior Editor,

rroberts@plos.org,

PLOS Biology

DATA POLICY:

Regardless of the method selected, please ensure that you provide the individual numerical values that underlie the summary data displayed in the following figure panels as they are essential for readers to assess your analysis and to reproduce it: Figs 2ACDE, 3BC, 4BCD, S2AB, S5AB, S6A, S8, S9, S10AB. NOTE: the numerical data provided should include all replicates AND the way in which the plotted mean and errors were derived (it should not present only the mean/average values).

We require the original, uncropped and minimally adjusted images supporting all blot and gel results reported in an article's figures or Supporting Information files. We will require these files before a manuscript can be accepted so please prepare and upload them now. Please carefully read our guidelines for how to prepare and upload this data: https://journals.plos.org/plosbiology/s/figures#loc-blot-and-gel-reporting-requirements 

DATA NOT SHOWN?

---

## [Editor Report · Decision Letter 3]

4 May 2022

Dear Dr Ziv,

On behalf of my colleagues and the Academic Editor, Joe Heitman, I'm pleased to say that we can in principle accept your Discovery Report "Multiple molecular events underlie stochastic switching between two heritable cell states in fungi" for publication in PLOS Biology, provided you address any remaining formatting and reporting issues. These will be detailed in an email that will follow this letter and that you will usually receive within 2-3 business days, during which time no action is required from you. Please note that we will not be able to formally accept your manuscript and schedule it for publication until you have completed any requested changes.

Sincerely,

Roli Roberts

Roland G Roberts, PhD 

Senior Editor 

PLOS Biology

rroberts@plos.org